# Sports Stars Brazil in children with autism spectrum disorder: A feasibility randomized controlled trial protocol

Amanda Cristina Fernandes[1], Deisiane Oliveira Souto[1], Ricardo R. de Sousa Junior[1], Georgina L. Clutterbuck[2], F. Virginia Wright[3], Mariane Gonçalves de Souza[1], Lidiane Francisca Borges Ferreira[4], Ana Amélia Cardoso Rodrigues[1], Ana Cristina R. Camargos[1], Hércules R. Leite[1] *

1 Graduate Program in Rehabilitation Sciences, School of Physical Education, Physical Therapy and Occupational Therapy, Universidade Federal de Minas Gerais, Belo Horizonte, Brazil, 2 School of Health and Rehabilitation Sciences, The University of Queensland, Brisbane, Queensland, Australia, 3 Holland Bloorview Kids Rehabilitation Hospital, and Department of Physical Therapy, University of Toronto, Toronto, Canada, 4 Graduate Program in Occupation Studies, School of Physical Education, Physiotherapy and Occupational Therapy, Federal University of Minas Gerais, Belo Horizonte, Brazil

* hercules@ufmg.br

**Data Availability Statement:** No datasets were generated or analysed during the current study. All

## Abstract

### Background

Autism Spectrum Disorder (ASD) children have lower levels of participation in recreational and sporting activities when compared to their peers. Participation has been defined based on the Family of Participation-Related Constructs (fPRC) which defines participation as including both attendance and involvement, with sense of self, preferences and activity competence related to a child's participation. Modified sports interventions such as Sports Stars can act on physical literacy and some of the fPRCs components. This study aims to assess the feasibility of the Sports Stars Brazil intervention for children with ASD.

### Methods

This study will be conducted with 36 participants with ASD aged 6 to 12 years old following the CONSORT for pilot and feasibility recommendation. Participants will be randomly allocated into two groups. Intervention group will receive eight, weekly Sports Stars sessions. Each session will include of sports-focused gross motor activity training, confidence building, sports-education and teamwork development. Study assessments will occur at baseline, immediately post-intervention and 20-weeks post-randomization. First, we will assess process feasibility measures: recruitment, assessment completion, adherence, adverse events and satisfaction. Second, we will investigate the scientific feasibility of the intervention by estimating the effect size and variance at the level of achievement sports-related activity and physical activity participation goals (Goal Attainment Scaling), activity competence (Ignite Challenge, Test of Gross Motor Development-second edition, Physical Literacy Profile Questionnaire, Pediatric Disability Assessment Inventory–Computer Adaptive Test—PEDI-CAT—mobility, 10×5 Sprint Test and Muscle Power Sprint Test), sense of self

relevant data from this study will be made available upon study completion.

**Funding:** The present study was supported by the Conselho Nacional de Desenvolvimento Científico e Tecnológico - CNPq (grant number 150010/2022-2 and 302010/2022-0), Pró-Reitoria de Pesquisa da UFMG (PRPq) and Coordenação de Aperfeiçoamento de Pessoal de Nível Superior (Coordination for the Advancement of Higher Education Personnel in the form of a PhD scholarship, grant number: APQ-00754-20).

**Competing interests:** The authors have declared that no competing interests exist.

(PEDI-CAT—responsibility), and overall participation at home, school and community, (Participation and Environment Measure for children and young people, PEM-CY).

## Discussion

The results of this feasibility study will inform which components are critical to planning and preparing a future RCT study, aiming to ensure that the RCT will be feasible, rigorous and justifiable.

## Trial registration

The trial was registered with the Brazilian Registry of Clinical Trials database (ID: RBR-9d5kyq4) on June 15, 2022.

## 1. Background

Participation in leisure-time physical activities prepares children for a physically active life and improves health and psychosocial outcomes [1]. Despite these benefits, children with disabilities are more likely to be sedentary and experience more barriers to participation in physical activities [2]. In particular, children with Autism Spectrum Disorder (ASD) have lower levels of physical activity compared to their peers [3–5]. ASD is a neurodevelopmental disorder that affects individuals' communication and social interaction, and includes repetitive or restricted behavior/s and interests [6, 7]. These children often experience impairments in their physical, social, cognitive and/or psychological competence, and these can pose significant barriers to their participation in sports [8]. Therefore, it is necessary to identify and promote interventions that increase participation in sports and recreational leisure-time physical activities in this population.

Participation is important as a core component of the evaluation of outcomes of children and adolescents with disabilities [9]. It is defined by the International Classification of Functioning, Disability and Health (ICF) as involvement in real-life situations [10]. More recently, the concept of participation has been extended to involve two essential components: attendance and involvement [11, 12]. The Family of Participation-Related Constructs (fPRC) defines participation as including both attendance and involvement, with sense of self, preferences and activity competence related to a child's participation [11, 12]. Activity competence is the ability to execute the activity being undertaken according to an expected standard, includes cognitive, physical and affective skills and abilities [11, 12], where can be measured: (1) abilities that the child can use in a daily environment, (2) ability of the child within a structured environment like that created for test-taking and (3) abilities the child uses in everyday settings [13, 14]. Attendance is characterized as "being present" and can be measured as frequency or diversity of activities that an individual participates [11, 12]. Involvement refers to the subjective experience of participation during attendance, including elements such as engagement, persistence, social bonding, and affection level [11, 12].

When considering participation in sports and recreational activities, activity competence can be associated with physical literacy. Physical literacy describes the skills that one needs to engage in enjoyable leisure-time physical activities throughout life, in four domains: 1) physical (e.g., locomotor skills); 2) psychological (e.g., motivation); 3) social (e.g., relationships); and 4) cognitive (e.g., content knowledge) [15–17]. The interaction between the fPRCs and physical literacy provides a greater understanding of the participation of children with ASD in physical and sports activities and provides guidance for future participation-based research.

Recently, Clutterbuck, Auld and Johnston [18] developed Sports Stars, a modified group sport intervention held in the community environment. Sports Stars has shown to be effective in improving preferred sports-related activity and physical activity participation (attendance and involvement) goals and activity competence for ambulant Australian children with cerebral palsy (CP) compared to usual care [18]. Furthermore, parents and therapists perceived that Sports Stars improved children's physical literacy across all domains, including the physical, social, psychological, and cognitive competence needed for sports participation, and that Sports Stars would be their intervention of choice for children with sports focused goals [19].

The prevalence and rate of diagnosis of children with ASD has been growing in recent years [20, 21]. According to the Brazilian Society of Pediatrics (SBP) [22], in Brazil, children with ASD frequently receive a late diagnosis and consequent delayed access to intervention, which might compromise their development [22]. There are significant advances in the international public policies for this population [20, 21]. However, in Brazil, laws were only recently introduced to guarantee the rights of people with ASD, for example to participate in sports and leisure activities (Laws: 12.764/2012–13.146/2015) [23, 24]. Despite the evidence suggesting that participation of people with ASD in leisure-time physical activities and group sports improves their socialization skills, communication, development of independence, motor skills and cardiovascular fitness, up to now, this is addressed by just one initiative by the Brazilian Government (entitled TEAtivo) [25–28]. Larger efforts are needed to developed appropriate interventions, such as Sports Stars, to improve physical activity levels and promote participation in sports and physical recreation for this population [29, 30]. However, while the effectiveness and feasibility of Sports Stars Brazil is under investigation in children with CP by our research group [31, 32], its feasibility in other conditions such as ASD is still unknown. This paper presents a feasibility Randomized clinical trial (RCT) protocol aiming to assess the feasibility of the Sports Stars Brazil intervention for children with ASD as articulated by Thabane et al. (2010) framework [33]. This framework encompasses that the aim of a feasibility study might be linked to one or more of the following four classifications: process, resources, management and scientific. In this study, we will focus on two feasibility classifications:

First, we will assess the process of feasibility that will determine the ability to enroll participants, the assessments completion rates, as well as adverse effects, satisfaction and adherence. To this purpose, a priori success criteria will be established when appropriated. Second, we will evaluate the scientific feasibility for estimating the effect size and variance of nine outcomes aligned with the fPRCs components.

## 2. Methods and analysis

### Study design

This will be a prospectively registered, open, two arm, pilot RCT. This manuscript was written in accordance with the SPIRIT (Standard Protocol Items: Recommendations for Interventional Trials) guidelines [34] (see S1 Checklist). Forthcoming publication of trial results will be reported according to reporting standards for pilot and feasibilities studies (i.e., Consolidated Standards of Reporting Trials—CONSORT), (see S2 Checklist) [35]. The schedule with assessment at different points in time is shown in Fig 1.

SPIRIT schedule of assessments at different time points

### Study setting

The study will be conducted in open spaces and sports courts/facilities of a university in the city of Belo Horizonte, Brazil.

| | STUDY PERIOD | | | | | | | | | | | | | |
|---|---|---|---|---|---|---|---|---|---|---|---|---|---|---|
| | Enrollment | Baseline assessments | Allocation | Intervention | | | | | | | | Post Intervention | Close -out |
| TIMEPOINT | -t₁ | t₀ | | t₁ | t₂ | t₃ | t₄ | t₅ | t₆ | t₇ | t₈ | 8 weeks | 12 weeks |
| **ENROLMENT:** | | | | | | | | | | | | | |
| **Eligibillity screen** | x | | | | | | | | | | | | |
| **Informed consente** | x | | | | | | | | | | | | |
| **Allocation** | | | x | | | | | | | | | | |
| **INTERVENTIONS:** | | | | | | | | | | | | | |
| Group Sports Stars | | | | x | x | x | x | x | x | x | x | | |
| Control Group | | | | | | | | | | | | | |
| **ASSESSMENTS:** | | | | | | | | | | | | | |
| ACSF-SC | x | | | | | | | | | | | | |
| Sensory Profile Questionaires | | x | | | | | | | | | | | |
| PEM-CY | | x | | | | | | | | | | x | x |
| GAS | | x | | | | | | | | | | x | x |
| Ignite Challenge | | x | | | | | | | | | | x | x |
| PLPQ | | x | | | | | | | | | | x | x |
| PEDI CAT + ASD | | x | | | | | | | | | | x | x |
| TGMD-2 | | x | | | | | | | | | | x | x |
| MPST | | x | | | | | | | | | | x | x |
| 10x5 ST | | x | | | | | | | | | | x | x |

**Fig 1. Time schedule of enrolment, interventions, and assessments on participant outcome inspired by the SPIRIT 2013 reporting guidelines [34].** Autism classification system of functioning: social communication—ACSF: SC; Participation and Environment Measure for children and young people -PEM-CY; GAS—Goal Attainment Scaling; Physical Literacy Profile Questionnaire- PLPQ; Pediatric Disability Assessment Inventory–Computer Adaptive Test—PEDI-CAT; Test of Gross Motor Development-second edition–TGMD-2; 10×5 Sprint Test -10x5 ST and Muscle Power Sprint Test–MPST.

## Eligibility criteria

Inclusion Criteria

- Children aged 6 to 12 years at the beginning of the intervention;

- Diagnosed with ASD according to medical report, and classified at support levels I or II by the Autism classification system of functioning: social communication ACSF:SC [36];

- Both genders.

  Exclusion Criteria

- Participants with cognitive, behavioral, or clinical limitations that prevent them from following instructions and safely participating in group physical activity;

- Who have undergone surgery or fractures in the last 6 months.

## Procedures

Children with ASD will be recruited through advertisement, radio spots, social media networks, as well as recruited by convenience from public or philanthropic institutions and private rehabilitation clinics in Belo Horizonte, Brazil. All eligible individuals will receive clarifications regarding the objectives of the study and will sign a statement of informed consent prior to participation. Participants characteristics will be collected. Finally, two blinded researchers will collect the outcome data at baseline, 8 weeks after randomization, and the follow-up evaluation. All assessments will be repeated by the same assessors.

## Data collection

**Characteristics of the participants.** Participants' age, sex, classification on the ACSF:SC [36], and the Abbreviated Sensory Profile 2 [37] will be collected after signing the informed consent and assent form.

**Autism classification system of functioning: Social communication (ACSF: SC).** The ACSF:SC classifies the level of communication skills used by children with ASD [36]. Classification is divided into five levels that distinguish individuals' social communication skills according to social needs and goals. The ACSF:SC was properly translated for Brazilian children and young people with ASD and will be applied by a trained assessor [38].

**Sensory profile questionnaires.** Sensory profiles of children in this study will be captured by the Abbreviated Sensory Profile 2. This questionnaire consists of 34 items which are answered by the caregiver and describes children's responses to various sensory experiences [39]. Responses should consider how often (always, often, occasionally, rarely, never) the behaviors occur, and scores are given from 1 (always) to 5 (never). Lower scores indicate greater severity of sensory problems.

**Outcomes.** *Process and criteria for feasibility*. The process feasibility measures will include: Recruitment rates, number of patients who completed the study, as well as adherence, adverse events and satisfaction. Child's adherence to the program will be evaluated through document analysis that will determine the number of children who started and completed the intervention. Interventionist reports on children's behavior during the sessions and possible adverse events observed by them will also be documented. To analyze the general satisfaction of the participants, a semi-structured questionnaire adapted from Feitosa et al. [40] will be used (S1 File). This questionnaire will contain questions related to the format of the intervention, for example, satisfaction with the intervention schedule. This questionnaire also addresses the child's interaction with the therapist, achievement of your prioritized goals. The completion of the questionnaire will be performed by the participants after the last day of the intervention.

The criteria or success for feasibility based on previous studies [41, 42] will include: ≥10% response rate from all eligible participants; and ≥80% of the participants successfully complete the study. (i.e. completed baseline, immediate and post follow up). Given the nature of the other feasibility measures they will not have cut-off criteria to determine the feasibility of performing a complete RCT. Instead, their results will be reported descriptively and will be used to determine the suitability of a full RCT combined with the others specific criteria mentioned.

This protocol study will follow one of the recommendations: not feasible, feasible with minor modifications, or feasible with closing monitoring [33].

Scientific feasibility. Preferred sports-related activity and physical activity participation goals

1. Activity and physical activity participation goals will be evaluated by the Goal Attainment Scaling–GAS [43, 44]. GAS is patient centred and has been widely used to quantify the achievement or fulfillment of goals previously defined in an intervention program [43–46]. In this study, goals will be selected by caregivers. In order to increase responsiveness, validity and reliability, three goals that can be affected by the treatment will be identified [47]. The first refers to performance of a sport-related motor activity, the second and third refers to attendance and involvement in leisure-time physical activities during sports or physical recreation participation (see example in Table 1). All GAS' will be shared among caregivers and interventionists at baseline by another non-blinded assessor.

*Activity Competence and sense of self*

1. ***Ignite Challenge*:** Measures accuracy and speed of locomotor and object control skills necessary for sports and active recreation activities [48]. It is a 13-item measure that can be used in children with ASD classified at ACSF levels I and II. The instrument total raw score is converted in percentage points. The higher the score, the better the performance. It can

**Table 1. Goal attainment scale examples.**

| Score | Meaning | Sports-related activity example | Attendance of physical activity participation | Involvement of physical activity participation |
|-------|---------|--------------------------------|-----------------------------------------------|------------------------------------------------|
| | | *Parent's goal: To dribble a soccer ball with control along the soccer field* | *Parent's goal: To participate in soccer classes at least twice in a week* | *Parent's goal: To be engaged at least 30 minutes when participates in school's sports activities of one-hour duration* |
| -2 | Current level | Dribbles a soccer ball during 4 meters of the soccer field without lose control of the ball | Does not participate in soccer classes in a week | Is not engaged at all during school's sports activities |
| -1 | Less than expected | Dribbles a soccer ball during 6 meters of the soccer field without lose control of the ball | Participate in soccer classes one day in a week | Is engaged in 15 minutes during school's sports activities |
| 0 | Expected to achieve | Dribbles a soccer ball during 10 meters of the soccer field without lose control of the ball | Participate in soccer classes two days in a week | Is engaged in 30 minutes during school's sports activities |
| +1 | More than expected | Dribbles a soccer ball during 12 meters of the soccer field without lose control of the ball | Participate in soccer classes three days in a week | Is engaged in 45 minutes during school's sports activities |
| +2 | Much more than expected | Dribbles a soccer ball during 14 meters of the soccer field without lose control of the ball | Participate in soccer classes four days in a week | Is engaged in 60 minutes during school's sports activities |

include "picture cards" for each test item to supplement evaluators' demonstrations and improve understanding of each test item [49–51]. The *Ignite Challenge* has demonstrated excellent inter-rater (ICC = 0.91 (95% = 0.93, 0.99), intra-rater (ICC = 0.96, 95% = 0.90, 0.98) and rest-test (ICC = 0.91, 95% = 0.84, 0.95) reliability in Australian children with ASD (n = 47) [51].

2. **Test of Gross Motor Development-2—TGMD-2:** Is used in children up three to 10 years, it assesses accuracy and quality of 12 fundamental motor skills, six of which are locomotor skills and six are object control skills. The total scores of each subtest are summed and represented as raw scores, which can be converted into motor quotients [52]. The TGMD-2 was validated and reliable for typical Brazilian children in the study by Valentini et al. [53] and has been used in children with ASD [54, 55]. Furthermore, it has been seen as a responsive measure in the Australian Sports Stars in children with CP [18]. Reliability of the TGMD-2 for children with ASD is currently being evaluated by our research group (ongoing study).

3. **10×5 Sprint Test–(10×5ST) and Muscle Power Sprint Test (MPST):** The 10×5ST assesses agility [56, 57]. In the 10×5ST, participants need to run 5 m separated by 2 cones, 10 times continuously, making turns in the cones that mark the end of the five meters [57]. The 10×5ST has excellent inter-observer and test-retest reliability (ICC = 1.00 and 0.97) and good reported construction validity. An increase in agility of 3.2 seconds is considered a real change [54]. Participants muscle power will be evaluated by the Muscle Power Sprint Test–MPST [54]. The MPST measures muscle power by asking the participant to run as fast as possible for 15 m, 6 times, with a 10 second interval between each sprint. The MPST has high inter-observer and test-retest reliability (ICC = 0.97 and 0.99) for ambulant children with CP [56]. Reliability of the MPST and 10x5ST for children with ASD is currently being evaluated by our research group (ongoing study).

4. **Pediatric Evaluation of Disability Inventory—Computer Adaptive—PEDI-CAT ASD:** Will be used to assess the domains of mobility (activity competence) and sense of self (responsibility) [58]. In the PEDI-CAT domain mobility the four-point scores are based on

different levels of difficulty. The responsibility domain ranks items on a five-point scale, describing the division of responsibility between the caregiver and the child in managing complex, multi-step life tasks. The PEDI-CAT was adapted and presented reliability to be used by children with autism [59]. It was translated and culturally adapted for the Brazilian population aged 0–21 years [60].

5. **Physical Literacy Profile Questionnaire–PLPQ:** Assesses the components of physical literacy (physical, cognitive, social and psychological) of children [61]. The questions are related to performance and satisfaction in the physical, social, psychological and cognitive competencies. Each item assesses performance on a scale of 0 to 2 and each item assesses the child/adolescent/young adult's performance satisfaction on a scale of 1–10 points. The higher the score, the higher the level of performance and satisfaction. The QPAF is in the process of being validated by our research group, preliminary data showed good test-retest reliability for children with CP (ICC = 0.84; IC [95%]: 0.74–0.91) (data not published yet) [61].

*Overall Participation at home, school and community*

6. **Participation and Environment Measure for Children and Youth—PEM-CY:** Will be used to assess participation at home, school and community [62, 63]. The measure identifies attendance and involvement of children in activities carried out at home, at school and in the community, as well as the characteristics of these environments that influence participation in six parts: frequency of participation, involvement of participation, desire to change participation, support for the environment, support for the environment and environmental resources [64, 65]. The PEM-CY has moderate to good internal consistency and test-retest reliability indices [62] and was translated and culturally adapted for the Brazilian population [58]. The PEM-CY it has been used as an outcome measure to assess the effects of participation interventions on individuals with ASD [66, 67].

*Randomization and blinding.* Children (n = 38) will be randomized into 2 groups (Sports Stars Brazil intervention and control group). Randomization will occur when 8 to 10 the child has been recruited and the allocation ratio will be 1:1. The randomization will be performed using a computer-generated random sequence to ensure equal allocation to each group. This sequence will be used to randomize children into the immediate group, or the control group. A new sequence will be used for each subgroup randomization until 36 children are allocated, or no further participants can be recruited. All assessments will be performed before the allocation of each subgroup. Thus, the use of the block randomization method is unlikely to increase the probability of identifying the allocation of participants.

An independent researcher, not involved in recruitment or data collection and without direct contact with those involved in this research, will perform all randomization steps. The randomization process and allocation of participants will be supervised by the independent investigator. Due to the intervention characteristics of this study, it is not possible to blind participants and interventional therapists to group allocation. In order to minimize bias, the children and their caregivers will be instructed to not tell the assessors which group they are in until after all their baseline assessments were completed. Furthermore, all the two blinded assessors will be asked to indicate if they know which group (control or intervention, and if so to cite the source of unblinding). This will permit to report the success of blinding. The statistician will be blinded to the group allocation until the completion of the analyses.

The possible contamination of information between the groups will also be evaluated. For this, the following questions will be asked. 1) Have you talked to other participants in this study about the intervention they are receiving? 2) If so, did your attitude towards the intervention change after talking to one of the participants in the other group? 3) Did you have any

changes related to physical activity after the first contact with our project? 4)Are any of the participants in the other group aware of the type of intervention you were receiving in this study?

## Interventions

**Sports Stars Brazil group.** Sports Stars will be conducted in small groups of four or five children aged 6–12 years old, with ASD and ACSF-SC level I-II, led by one physical therapist (the same in all Sports Stars groups) with the assistance of Occupational Therapy and Physical Education undergraduate and graduate students. Each session will consist of the following activities: arrival of participants (10 minutes); warm up (5 minutes), locomotor skill training, e.g. running and jumping activities (15 minutes); object control skill training; e.g. dribbling, discus, bouncing, catching (15 minutes); modified sports game (10 minutes); and a cool down (5 minutes) [18, 31]. Participants will receive eight, one-hour, weekly sessions of Sports Stars Brazil, with two weeks focusing on each of the following popular Brazilian sports: soccer, handball, basketball and athletics, per the published protocol for children with CP [31]. The structure main components of Sports Stars Brazil, as well as the strategies for supporting the autistic children during the sessions are detailed in Fig 2. Along the program the complexity of the task is graded, aiming to improve the child's performance and developing activity competence in each of the physical literacy domains (i.e., physical, cognitive, psychological and social skills). Standard descriptors are used to guide each child's progress, as detailed in the Sports Stars Session Plan examples (see S2 File)".

Since children with ASD may present with repetitive and stereotyped behaviors [7], support strategies based on the guidelines of the I CAN Develop Physical Literacy, carried out in partnership by the Physical Literacy Programs (PISE) and Canucks Autism Network (CAN) [68] will be used to facilitate participation during Sports Stars [69]. Specifically, during the reception, a visual schedule will be used to introduce the day's activities, providing predictability of

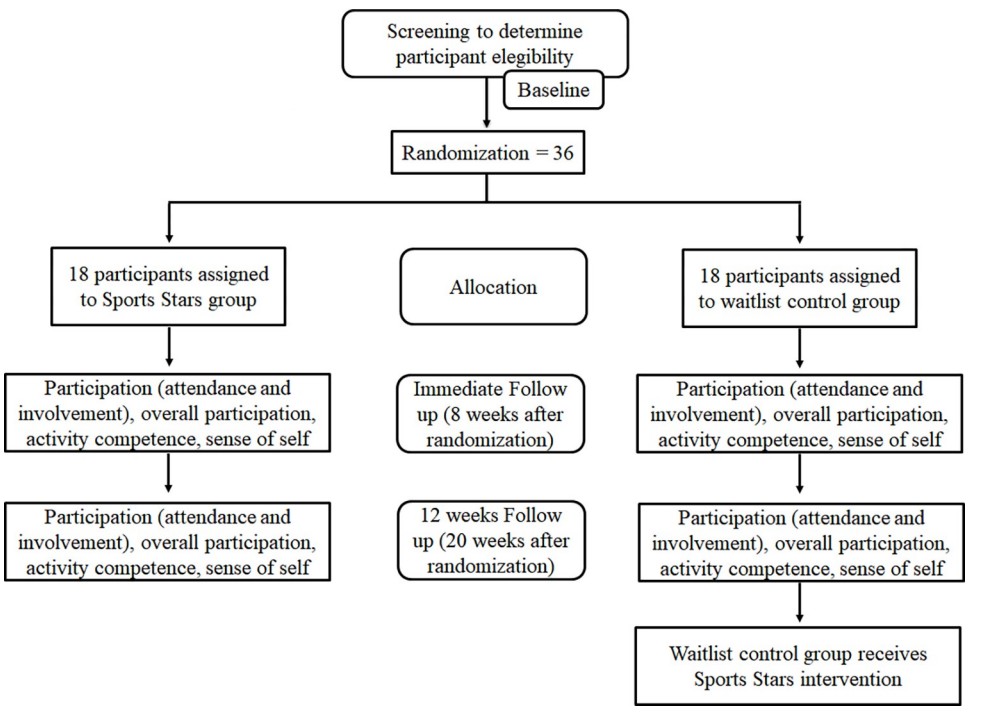

**Fig 2. Structure and main components of the Sports Stars intervention.**

the session. During the sessions strategies such as giving feedback, informing about rest time, time to hydrate and time to relax will be also used, Fig 2. All these strategies will be documented through the study.

Structure of the Sports Stars intervention and the supporting strategies applied.

**Usual care or control group.** Participants in the control group will receive their standard care, including maintaining existing occupational therapy and/or physical therapy intervention programs. Children with ASD in Brazil are expected to receive 1 to 2 sessions of occupational therapy (but not physical therapy) per week in public or private clinics. Both interventions provide individualized treatment plans tailored to the needs of each child. Generally speaking, physical therapy provides a general exercise program that involves gross motor training, muscle strengthening, and balance and coordination training. Occupational therapy usually involves sensory integration, neuropsychomotor development and participation in occupations (activities of daily living and education). The activities done by participants in the control group and the treatment adherence will be registered in a daily activity (see S3 File). To ensure equal access to this intervention, after they have participated in their post-intervention and follow-up assessments, participants in the control group will be invited to receive the 8-week Sports Stars Brazil intervention, but no outcomes will be collected.

## Participant timeline

The flowchart summarizing the experimental procedures and participants is displayed in Fig 3.

## Statistical methods

**Sample size rationale.** This study is designed to investigate the feasibility of conducting a future RCT to evaluate the effectiveness of Sports Stars and to build decision-making processes to guide the execution of a larger study, particularly concerning satisfaction, adherence. Despite of no sample size calculation is necessary in a feasibility trial, 15–20 subjects by group are suggested to determine if enrollment is sufficient to progress to a full RCT [33]. Our aim is to identify 36 potential participants to reach our target sample (18 per group) [70].

*Data analysis.* For process feasibility, participant recruitment and demographic/clinical data will be reported as means and standard deviations for continuous parametric data, medians and ranges for continuous non-parametric data, and frequencies and percentages for categorical data as per CONSORT recommendations [35]. Success criteria for feasibility will be reported descriptive and narratively, with accompanying 95% Confidence Interval (CI), when appropriate.

For scientific feasibility, as the objective of the study is to inform a definitive RCT, the analysis will focus on estimating treatment effect size and variance. No statistical significance tests

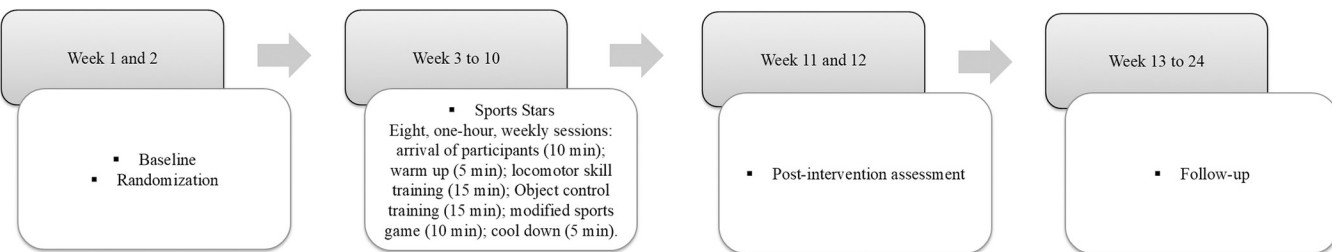

**Fig 3. Study flowchart.** TGMD-2: Test of Gross Motor Development-2; GAS: Goal Attainment Scaling; 10×5ST 3: 10×5 Sprint Test; MPST: Muscle Power Sprint Test; PEDI-CAT ASD: Pediatric Evaluation of Disability Inventory—Computer Adaptive; PLPQ: Physical Literacy Profile Questionnaire; PEM-CY: Participation and Environment Measure for Children and Youth.

(p value) or hypotheses regarding the effectiveness of the treatment will be performed. Treatment effects for secondary outcome measures will be presented as means, SD and CIs [35]. Considering a normal data distribution, effects sizes for each scientific feasibility outcome will be collected pre-intervention, immediately post intervention and at 12 weeks post intervention, using the following equation:

$$Cohen\acute{s}\ D = \frac{Mean\ pre-post\ change\ (treatment) - Mean\ pre-post\ change\ (control)}{Standard\ deviation\ (pooled)}$$

The following thresholds will be considered for interpretation of effect size: small (0.20 0.49), medium (0.50–0.79) and large (>0.80). High scores indicate better outcomes and positive effect sizes suggest benefit from Sports Stars over the control group [71]. The 95% CI will also be reported.

### Plan for supervision and monitoring

The study will be conducted and monitored by the lead investigator (HRL) under the supervision of the author (ACF), with assistance of the research team. All the ethical principles as provided by Declaration of Helsinki will be followed by all the members of this research.

### Ethical consideration

This study received approval from the ethics committee of the university (certificate number:55151222.4.0000.5149) and was prospectively registered in the Brazilian Registry of Clinical Trials: RBR-9d5kyq4. Written consent will be obtained from parents or caregivers of each participant. In the same way, children will only participate in the study by signing the free and informed assent term. Participants' information will be coded to preserve their identity. On completion of the study, data will be analyzed and tabulated and a final study report will be prepared. The researchers will take all appropriate and customary steps to ensure that the data remain safeguarded and that the privacy and confidentiality of the participants are maintained. Protocol modifications will be reported to the Institutional Review Board and to the trial registry.

### Data integrity and management plan

Research data will be collected by two research assistants who will be trained to collect and manage it. Participant identifiers (including name, address, contact information and other personal information) will be removed from the survey data and stored in another file. The data will be entered into Microsoft Excel and SPSS. Survey data will be monitored weekly and any errors in input will be identified (if any) and amended. Consent forms will be stored in the office of the School of Physical Education, Physiotherapy and Occupational Therapy at the Federal University of Minas Gerais along with other research data files.

## 3. Discussion

### Potential impact and significance of the study

This study presents a protocol of a feasibility RCT of Sports Stars Brazil in children with ASD compared to a control group. The main objective of the randomized pilot or feasibility testing is to assess the feasibility of drive the definitive future RCT. The increasing prevalence of ASD indicates that a greater number of pragmatic interventions with capacity for rapid upscale are needed to improve participation in sport and physical recreation for this population. This proposed pilot RCT will provide relevant information (e.g., adherence, adverse effects and

satisfaction) to plan a full RCT that can assess the effectiveness of Sports Stars Brazil for children with ASD.

## Strengths and weaknesses of the study

This feasibility study has a number of strengths. The Sports Stars program has been tailored to meet the needs of children with ASD, such as to promote participation in meaningful leisure activities, and delivery in group format to motivate children to play together. Furthermore, the secondary outcomes address important elements of Participation and its related constructs across the fPRC. The study protocol was based CONSORT to randomized pilot and feasibility trials [35], increasing the likelihood that crucial study design elements have been covered. In addition, all therapists will receive ongoing expert (G.C and VW are the researchers who developed originally the intervention in Australia and the Ignite Challenge assessment, respectively) consultation throughout the study.

This feasibility study will exclude children classified over the ACSF-SC level II, which may limit the generalizability of findings A potential weakness is that many assessments relevant to sport and physical recreation lack psychometric information when used with children with ASD. However, for measures which do not currently have published reliability data, further evaluation is currently underway by our research group and can be excluded from this study if its psychometric properties are found not to be satisfactory. Furthermore, it is not possible to blind the children and the interventionists. Finally, no specific instrument will be applied to assess possible cognitive alterations of the participants. Only subjective assessment of cognitive functioning will be performed during screening or through parents' reports.

## Contribution to physical therapy profession

To the best of our knowledge, this will be the first feasibility RCT study to investigate a practitioner-led, peer-group sports intervention for children with ASD. This study is an important first step in establishing the feasibility of a new intervention to improve leisure-time physical activity participation for children with ASD. In conclusion, the findings from this feasibility RCT will further our understanding of an intervention that has the potential to address the functioning (i.e., impairments, activity limitation and participation restrictions) of children with ASD.

## Supporting information

**S1 Checklist. SPIRIT 2013 checklist: Recommended items to address in a clinical trial protocol and related documents.**
(DOC)

**S2 Checklist. CONSORT 2010 checklist of information to include when reporting a pilot or feasibility trial.**
(DOC)

**S1 File. Satisfaction questionnaire for the Sports Stars group.**
(DOCX)

**S2 File. Sports Stars Session Samples.**
(DOCX)

**S3 File. Daily activity.**
(DOCX)

**S4 File.**
(PDF)

**S5 File.**
(PDF)

## Author Contributions

**Conceptualization:** Georgina L. Clutterbuck, F. Virginia Wright.

**Data curation:** Amanda Cristina Fernandes, Deisiane Oliveira Souto, Ricardo R. de Sousa Junior, Mariane Gonçalves de Souza.

**Formal analysis:** Amanda Cristina Fernandes, Ricardo R. de Sousa Junior, Mariane Gonçalves de Souza.

**Methodology:** Amanda Cristina Fernandes, Ricardo R. de Sousa Junior, Mariane Gonçalves de Souza, Ana Amélia Cardoso Rodrigues.

**Project administration:** Georgina L. Clutterbuck, Hércules R. Leite.

**Supervision:** F. Virginia Wright, Hércules R. Leite.

**Validation:** F. Virginia Wright, Hércules R. Leite.

**Visualization:** F. Virginia Wright, Ana Cristina R. Camargos, Hércules R. Leite.

**Writing – original draft:** Amanda Cristina Fernandes, Deisiane Oliveira Souto, Georgina L. Clutterbuck, F. Virginia Wright, Ana Amélia Cardoso Rodrigues, Ana Cristina R. Camargos, Hércules R. Leite.

**Writing – review & editing:** Amanda Cristina Fernandes, Deisiane Oliveira Souto, Georgina L. Clutterbuck, F. Virginia Wright, Lidiane Francisca Borges Ferreira, Ana Amélia Cardoso Rodrigues, Ana Cristina R. Camargos, Hércules R. Leite.

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
