## [Decision Letter · Decision Letter 0]

15 Feb 2023

PONE-D-22-31855Effectiveness of Sports Stars Brazil in children with autism spectrum disorder: A randomized controlled trial protocol Sports Stars Brazil in children with autism spectrum disorder

PLOS ONE

Dear Dr. Souto,

Thank you for submitting your manuscript to PLOS ONE. After careful consideration, we feel that it has merit but does not fully meet PLOS ONE’s publication criteria as it currently stands. Therefore, we invite you to submit a revised version of the manuscript that addresses the points raised during the review process.

We look forward to receiving your revised manuscript.

Kind regards,

Zakir Abdu (Assistant Professor)

Academic Editor

PLOS ONE

Journal Requirements:

Additional Editor Comments (if provided):

Dear Author (s),

Thank you for submitting an important and scientific piece of work in the field of SRH Effectiveness of Sports Stars Brazil in children with autism spectrum disorder in PLOS ONE journal. With due reverence to our reviewers and valuable comments, the following points are additional ones found to be addressed, the manuscript writing and its flow are praiseworthy though :

Rewrite the abstract part as per the PLoSONE policy.Methodology part: The trial wil be reported following the Standard Protocol Items for Randomized Interventional …. (an English grammar problem)Clearly describe the sample size calculation with rationaleWhat do you think about information contamination?Who will be given the intervention?Briefly explain the treatment protocol

Reviewers' comments:

Reviewer's Responses to Questions

**Comments to the Author**

1. Does the manuscript provide a valid rationale for the proposed study, with clearly identified and justified research questions?

Reviewer #1: Partly

Reviewer #2: Yes

Reviewer #3: Yes

2. Is the protocol technically sound and planned in a manner that will lead to a meaningful outcome and allow testing the stated hypotheses?

Reviewer #1: Partly

Reviewer #2: Yes

Reviewer #3: Partly

3. Is the methodology feasible and described in sufficient detail to allow the work to be replicable?

Reviewer #1: No

Reviewer #2: Yes

Reviewer #3: Yes

4. Have the authors described where all data underlying the findings will be made available when the study is complete?

Reviewer #1: Yes

Reviewer #2: Yes

Reviewer #3: Yes

5. Is the manuscript presented in an intelligible fashion and written in standard English?

Reviewer #1: Yes

Reviewer #2: Yes

Reviewer #3: Yes

6. Review Comments to the Author

You may also provide optional suggestions and comments to authors that they might find helpful in planning their study.

Reviewer #1: Comments to authors

The paper aims to assess Effectiveness of Sports Stars Brazil in children with autism spectrum disorder: A randomized controlled trial protocol Sports Stars Brazil in children with autism spectrum disorder

The paper is relevant in its discipline. Although the paper is well written, the authors need to consider some issues as highlighted below.

Title

Should you use the word “effectiveness” or “effect”?

Abstract

your abstract is well written but it would be better to include the Background which should include:

• Case (context)

• Gap (Punch Line)

Background

This section should give a summary of the main issue of the study. It is better to include the magnitude of the problem from globe to local, efforts made by the government or other concerned body to address the issue and the gap to be filled by this study.

Methods

Line 1 “an is” should be removed.

Line 2 wil should be changed to will.

The sampling procedure – in your document “procedure” is not clearly written. As this is a randomized controlled trial, how are going to select the participants and randomize??

Is it individual randomization? How are you going to minimize information contamination as this is a kind of behavior intervention??

These issues should be clearly stated in the sampling procedure?

Your sample size calculation is also not clear. Even if there is no previous study to estimate the effect size, you could have used the conventional effect size of small, medium or high and calculated your sample size.

Data analysis

“Differences between groups will be calculated using unpaired t tests”. Better to say independent t-test. There is not unpaired t- test in statistics. Paired - t-test or independent t- test.

“Differences between groups over time will be analyzed using repeated measured analysis of variance followed by Tukey post hoc test for parametric data”.

ANOVA is used for the crude analysis of mean difference between groups. It does not show the difference in difference. So it is advisable to use appropriate statistical models to determine the difference overtime between the groups.

Ethical consideration

“All participants signed the informed consent form”. But your study population is under the legal age to give informed consent.

From whom are you going to obtain consent? Assent??

Reviewer #2: This is a thorough and well written research proposal for pre-registration as a RCT. I only have a slight concern regarding the sample size. A priori power analysis for a 2x2 mixed factorial ANOVA assuming a medium effect size, 80% power, alpha of .05, and correlation between repeated measures of .50 calls for 34 total participants. The authors are assuming 20% dropout, so I wonder if a larger sample size should be required. I realize that if larger effect sizes than f = .25 and/or stronger correlations between repeated measures than .50 are observed, then 80% power could be maintained with a smaller sample, but these are unknowns. I wonder if the authors should justify their assumptions more clearly or perhaps target a larger sample. Otherwise, I am simply noting a few typos I saw:

p. 12, end of first paragraph typo (invertion > intervention)

p. 12, near the bottom typo (unbinding > unblinding)

Fig 1. (enrolment > enrollment)

Reviewer #3: COMMENTS – Manuscript PONE-D-22-31855

Title: “Effectiveness of Sports Stars Brazil in children with autism spectrum disorder: A randomized controlled trial protocol Sports Stars Brazil in children with autism spectrum disorder”

Important note: This review pertains only to ‘statistical aspects’ of the study and so ‘clinical aspects’ [like medical importance, relevance of the study, ‘clinical significance and implication(s)’ of the whole study, etc.] are to be evaluated [should be assessed] separately/independently. Further please note that any ‘statistical review’ is generally done under the assumption that (such) study specific methodological [as well as execution] issues are perfectly taken care of by the investigator(s). This review is not an exception to that and so does not cover clinical aspects {however, seldom comments are made only if those issues are intimately / scientifically related & intermingle with ‘statistical aspects’ of the study}. Agreed that ‘statistical methods’ are used as just tools here, however, they are vital part of methodology [and so should be given due importance]. I look at the manuscript in/with statistical view point, other reviewer(s) look(s) at it with different angle so that in totality the review is very comprehensive. However, there should be efforts from authors side to improve (may be by taking clues from reviewer’s comments). Therefore, please do not limit the revision only (with respect) to comments made here.

COMMENTS: Overall though there are not many flaws in the manuscript, I have few concerns. First and foremost {very serious} is about the ‘sample Size’ section. You stated that “due to the lack of suitable research to determine effect size, the sample size calculation was performed ---” which may be true. But even for medium effect size, according to table-2 on page 158 of Cohen’s paper “A power primer” in Psychological Bulletin, 1992, vol.:112, pp 155-159 [which is a sort of summary of the excellent book by Jacob Cohen titled ‘Statistical power analysis for the behavioral sciences’, Academic Press, 1977, New York] even for medium effect size you need n=64 per group (type-I error=0.05, power=80%). Therefore, what is you stated [“the sample size was performed considering a parallel group design, with no weighted patient preference, power of 0.81, type I error rate of 0.05, means method of analysis, and a loss up to 20% over time, a total of 36 participants will be required for the study (18 per group)] then implies that there is certainly some very serious error in your calculations. Kindly check (and modify the protocol manuscript accordingly).

Note that ‘How the required minimum sample size for this study was determined is nevertheless a very-very important question [one of the important items in CONSORT checklist, item 7a] for any type of study (clinical trial or else) which needs to be discussed in adequate details {including assumptions made at the time of estimation, power of the study, software used, etc.}’. Moreover, in ‘Strengths and Weakenesses of the Study’ section you say “The sample size was calculated to provide the appropriate statistical power to detect precise between-group differences for the primary outcomes”. I very doubtful about this statement. Please re-check. Otherwise, this small sample size is acceptable only for ‘pilot’ study (but if considered as pilot study, ‘pilot’ word should appear in title).

I even doubt if just saying that (refer to ‘Methods and Analysis - Study Design’ section) that “The results will be further reported following the Consolidated Standards of Reporting Trials (CONSORT)” is sufficient? I guess, not (because the important phrase ‘CONSORT’ appears only once [though most of the important items are discussed/included] in this manuscript). It is well-known that while reporting [findings from and even planning] ‘Clinical Trial’ one should follow CONSORT guidelines [since your article type is ‘Clinical Trial’, you are supposed to cover important CONSORT items in the report or even in the ‘Protocol’].

In ‘Data Analysis’ section since you stated that “Differences between groups will be calculated using unpaired t tests”, please note that, [especially because “the sample size calculation was performed applying a probabilistic model for data for trials from our primary outcome, the Goal Attainment Scaling (GAS) endpoint”], please note that though the measures/tools used are appropriate, most of them yield data that are in [at the most] ‘ordinal’ level of measurement [and not in ratio level of measurement for sure {as the score two times higher does not indicate presence of that parameter/phenomenon as double (for example, a Visual Analogue Scales VAS score or say ‘depression’ score)}]. Then application of suitable non-parametric test(s) is/are indicated/advisable [even if distribution may be ‘Gaussian’ (also called ‘normal’)]. Agreed that there is/are no non-parametric test(s)/technique(s) available to be used as alternative in all situation(s) [suitable / most desired/applicable], but should be used whenever/wherever they are available. This [‘ordinal’ level of measurement of data of primary outcome, the Goal Attainment Scaling (GAS) endpoint] also has implications in estimation of required sample size [it is sometimes suggested to inflate the sample size by 20-25% while dealing with ‘ordinal’ level of measurement data].

As pointed out in ‘important note’ above “This review pertains only to ‘statistical aspects’ of the study and so ‘clinical aspects’ should be assessed separately/independently [one should carefully consider/look at the clinical implications of the study]. In my opinion, to rescue this article (which is quite possible), some amount of re-vision (re-drafting) may be needed. However, please do not limit the revision only (with respect) to comments made here. The respected ‘Editor’ may consider accepting if found ‘clinical implications’ (of this study) valuable. ‘Major revision’ is recommended [assuming that the respected editor would to like to give chance of improvement of the manuscript as in my opinion, the study has potential].

7. PLOS authors have the option to publish the peer review history of their article (what does this mean?). If published, this will include your full peer review and any attached files.

Reviewer #1: **Yes: **Dereje Tsegaye Ph.D

Reviewer #2: **Yes: **Daniel M. Smith

Reviewer #3: No

---

## [Author Response · Author response to Decision Letter 0]

27 Mar 2023

PLOS ONE

Dear Editor,

First of all, we thank the you and reviewers for the comments and suggestions regarding the manuscript: “Sports Stars Brazil in children with autism spectrum disorder: A pilot randomised controlled trial protocol”. We have addressed all the comments and suggestions in this letter and revised the manuscript accordingly. All the modifications are highlighted in red in the new version of the manuscript. We would like to inform that, in order to provide the manuscript’s adjustments suggested by the reviewers, a new author collaborated on this new version of the manuscript and was added as a co-author.

Best Regards,

Hércules Ribeiro Leite, Ph.D.

Universidade Federal de Minas Gerais

E-mail: (hercules@ufmg.br).

EDITORIAL REQUESTS

Editor, Comment 1

Rewrite the abstract part as per the PLoS ONE policy. 

Authors’ answer: Done as requested. The new edits followed the Plos One recommendations.

Editor, Comment 2

Methodology part: The trial will be reported following the Standard Protocol Items for Randomized Interventional …. (an English grammar problem) 

Response: Done as requested.

Editor, Comment 3

Clearly describe the sample size calculation with rationale

Response: We acknowledge that the sample size calculation was the weaknesses of our study, as stated by the editor and all the three reviewers. We have previously based our sample size calculation on a previous study looking at the Goal Attainment Scaling (GAS) outcome. However, considering all the queries raised and the reviewer’s suggestion, we decided to reformulated our paper and aims, where a pilot RCT design have been reported throughout the updated manuscript. 

Pages 17-18, lines 363 to 371: “The sample size calculation was conducted using four main characteristics: significance level, power, an estimate of group difference and standard deviation.67 As there are no published studies on the effectiveness of Sports Stars intervention in children with ASD, to the ability to estimate group differences and variances is limited. It is known that one of the main goals of a pilot study is to give information on sample size for a future RCT.68 Therefore, while no sample size calculation is necessary in a pilot trial, 15-20 subjects by group are suggested to estimate effects from pilot studies and determine if enrollment is sufficient to progress to a full RCT.69 Our aim is to enroll 18 participants per group as the minimum sample size.70 “

Editor, Comment 4

What do you think about information contamination?

Response: To assess information contamination between groups, the following questions will be asked:

Page 15, lines 304 to 310: “The possible contamination of information between the groups will also be evaluated. For this, the following questions will be asked. 1) Have you talked to other participants in this study about the intervention they are receiving? 2) If so, did your attitude towards the intervention change after talking to one of the participants in the other group? 3) Did you have any changes related to physical activity after the first contact with our project? 4)Are any of the participants in the other group aware of the type of intervention you were receiving in this study?”.

Editor, Comment 5

Who will be given the intervention?

Response: Thanks for raising this query. The intervention will be led by one physical therapist (the same in all intervention groups) with the assistance of Occupational Therapy and Physical Education’s undergraduate and graduate students. In order to better clarify this issue, we have added new excerpts in the new version of the manuscript. 

Page 15, lines 313 to 316: “Sports Stars will be conducted in small groups of four or five children aged 6-12 years old, with ASD and ACSF-SC level I-II, led by one physical therapist (the same in all Sports Stars groups) with the assistance of Occupational Therapy and Physical Education undergraduate and graduate students.”

Editor, Comment 6

Briefly explain the treatment protocol

Response: Thanks for raising this query. We acknowledge that we have provided few details of our treatment protocol. In the new version of the manuscript, we added more detailed information and provided a supplementary material showing the protocol details. We have also built a new figure (Figure 2) to better summarize and clarify the Sports Stars protocol. 

Page 16, lines 323 to 329: “…The structure main components of Sports Stars Brazil, as well as the strategies for supporting the autistic children during the sessions are detailed in figure 2. Along the program the complexity of the task is graded, aiming to improve the child's performance and developing activity competence in each of the physical literacy domains (i.e., physical, cognitive, psychological and social skills). Standard descriptors are used to guide each child's progress, as detailed in the Sports Stars Session Plan examples (see Supporting information S4).”

REVIEWER: 1: 

Reviewer 1, Comment 1

Title

Should you use the word “effectiveness” or “effect”?

Response: Considering that we have changed the paper focus to a pilot RTC, we have excluded this term from the title and main text, please see below the new title: 

Page 1: “Sports Stars Brazil in children with autism spectrum disorder: A pilot randomized controlled trial protocol”.

Reviewer 1, Comment 2

Abstract

your abstract is well written but it would be better to include the Background which should include:

• Case (context)

• Gap (Punch Line)

Response: Done as requested.

Page 2, lines 3 to 10: “Autistic children have lower levels of participation in recreational and sporting activities when compared to their peers. Participation has been defined based on the Family of Participation-Related Constructs (fPRC) which defines participation as including both attendance and involvement, with sense of self, preferences and activity competence related to a child's participation. Modified sports interventions such as Sports Stars can act on physical literacy and some of the fPRCs components. This study aims to assess the feasibility of the Sports Stars Brazil intervention for children with ASD, informing the primary outcome measure and sample size for a full RCT.” 

Reviewer 1, Comment 3

Background

This section should give a summary of the main issue of the study. It is better to include the magnitude of the problem from globe to local, efforts made by the government or other concerned body to address the issue and the gap to be filled by this study.

Response: Done as requested.

Page 5-6, lines 93 to 106: “The prevalence and rate of diagnosis of children with ASD has been growing in recent years.20,21 According to the Brazilian Society of Pediatrics (SBP)22, in Brazil, children with ASD frequently receive a late diagnosis and consequent delayed access to intervention, which might compromise their development.22 There are significant advances in the international public policies for this population. 20,21 However, in Brazil, laws were only recently introduced to guarantee the rights of people with ASD, for example to participate in sports and leisure activities (Laws: 12.764/2012 - 13.146/2015).23,24 Despite the evidence suggesting that participation of people with ASD in leisure-time physical activities and group sports improves their socialization skills, communication, development of independence, motor skills and cardiovascular fitness, up to now, this is addressed by just one initiative by the Brazilian Government (entitled TEAtivo). 25,26,27,28 Larger efforts are needed to developed appropriate interventions, such as Sports Stars, to improve physical activity levels and promote participation in sports and physical recreation for this population.29,30”

Reviewer 1, Comment 4

Methods

Line 1 “an is” should be removed.

Line 2 wil should be changed to will.

Response: Done as requested.

Reviewer 1, Comment 5

The sampling procedure – in your document “procedure” is not clearly written. As this is a randomized controlled trial, how are going to select the participants and randomize??

Is it individual randomization? How are you going to minimize information contamination as this is a kind of behavior intervention??

These issues should be clearly stated in the sampling procedure?

Your sample size calculation is also not clear. Even if there is no previous study to estimate the effect size, you could have used the conventional effect size of small, medium or high and calculated your sample size.

Response: We appreciate your comment. We have added more information to the text in order to clarify the randomization and blinding procedure. About the sample calculation, as this was a point raised by the editor and the three reviewers, we decided to reformulate our article and chose to carry out a pilot RCT study. Therefore, we provide a new description of the sample size following the literature recommendation. Please see below the new excerpts added to new manuscript:

Pages 14-15, lines 284 to 310: “Children (n = 38) will be randomized into 2 groups (Sports Stars Brazil intervention and control group). Randomization will occur when 8 to 10 the child has been recruited and the allocation ratio will be 1:1. The randomization will be performed using a computer-generated random sequence to ensure equal allocation to each group. This sequence will be used to randomize children into the immediate group, or the control group. A new sequence will be used for each subgroup randomization until 36 children are allocated, or no further participants can be recruited. All assessments will be performed before the allocation of each subgroup. Thus, the use of the block randomization method is unlikely to increase the probability of identifying the allocation of participants.

An independent researcher, not involved in recruitment or data collection and without direct contact with those involved in this research, will perform all randomization steps. The randomization process and allocation of participants will be supervised by the independent investigator. Due to the intervention characteristics of this study, it is not possible to blind participants and interventional therapists to group allocation. In order to minimize bias, the children and their caregivers will be instructed to not tell the assessors which group they are in until after all their baseline assessments were completed. Furthermore, all the two blinded assessors will be asked to indicate if they know which group (control or intervention, and if so to cite the source of unblinding). This will permit to report the success of blinding. The statistician will be blinded to the group allocation until the completion of the analyses.

The possible contamination of information between the groups will also be evaluated. For this, the following questions will be asked. 1) Have you talked to other participants in this study about the intervention they are receiving? 2) If so, did your attitude towards the intervention change after talking to one of the participants in the other group? 3) Did you have any changes related to physical activity after the first contact with our project? 4)Are any of the participants in the other group aware of the type of intervention you were receiving in this study?”

Response: About the sample calculation, as this was a point raised by the editor and the three reviewers, we decided to reformulate our article and chose to carry out a pilot RCT study. Therefore, we provide a new description of the sample size following the literature recommendation. Please see information below highlighted in red in the manuscript text.

Pages 17-18, lines 363 to 371: “The sample size calculation was conducted using four main characteristics: significance level, power, an estimate of group difference and standard deviation.67 As there are no published studies on the effectiveness of Sports Stars intervention in children with ASD, to the ability to estimate group differences and variances is limited. It is known that one of the main goals of a pilot study is to give information on sample size for a future RCT.68 Therefore, while no sample size calculation is necessary in a pilot trial, 15-20 subjects by group are suggested to estimate effects from pilot studies and determine if enrollment is sufficient to progress to a full RCT.69 Our aim is to enroll 18 participants per group as the minimum sample size.70”

Reviewer 1, Comment 6

Data analysis

“Differences between groups will be calculated using unpaired t tests”. Better to say independent t-test. There is not unpaired t- test in statistics. Paired - t-test or independent t- test

ANOVA is used for the crude analysis of mean difference between groups. It does not show the difference in difference. So it is advisable to use appropriate statistical models to determine the difference overtime between the groups.

Response: We acknowledge that the statistical analysis and sample size calculation were the weaknesses of our study, as stated by the editor and all the three reviewers. Thus, considering all the queries raised and the reviewer’s suggestion, we decided to reformulate our paper, where a pilot RCT design have been reported accordingly throughout the updated manuscript. Consequently, we have also updated the statistical analysis to follow the new aims and purposes of the study.

Page 18, lines 373 to 389: “Demographic and clinical data will be reported as means and SDs for continuous parametric data, medians and ranges for continuous non-parametric data, and frequencies and percentages for categorical data. To assess feasibility (adherence, adverse effects and satisfaction), a descriptive data analysis will be implemented, with accompanying 95% Confidence Interval (CI).

Considering a normal data distribution, effects sizes for each secondary outcome will be calculated after 8 and 12 weeks postintervention, as follows:

The following thresholds will be considered for interpretation of effect size: small (0.20 0.49), medium (0.50–0.79) and large (>0.80). High scores indicate better

outcomes and positive effect sizes suggest benefit from Sports Stars over the control group.71 An intention-to-treat analysis will be used. Sample size calculations for the RCT will be made using the treatment effect size and variance estimates from the immediate postintervention change data for the selected outcome measure. Missing follow-up data will be addressed using pairwise deletion.”

Reviewer 1, Comment 7

Ethical consideration

“All participants signed the informed consent form”. But your study population is under the legal age to give informed consent.

From whom are you going to obtain consent? Assent??

Response: Written consent will be obtained from parents or caregivers of each participant. In the same way, written assent will be obtained from each child. This information has been added to the manuscript, as following”

Page 19, lines 499 to 403: “Written consent will be obtained from parents or caregivers of each participant. In the same way, written assent will be obtained from each child. Participants’ information will be coded to preserve their identity. On completion of the study, data will be analyzed and tabulated and a final study report will be prepared.”

REVIEWER: 2: 

Reviewer 2, Comment 1

This is a thorough and well written research proposal for pre-registration as a RCT. I only have a slight concern regarding the sample size. A priori power analysis for a 2x2 mixed factorial ANOVA assuming a medium effect size, 80% power, alpha of .05, and correlation between repeated measures of .50 calls for 34 total participants. The authors are assuming 20% dropout, so I wonder if a larger sample size should be required. I realize that if larger effect sizes than f = .25 and/or stronger correlations between repeated measures than .50 are observed, then 80% power could be maintained with a smaller sample, but these are unknowns. I wonder if the authors should justify their assumptions more clearly or perhaps target a larger sample.

Response: We acknowledge that the statistical analysis and sample size calculation were the weaknesses of our study, as stated by the editor and all the three reviewers. Thus, considering all the queries raised and the reviewer’s suggestion, we decided to reformulate our paper, where a pilot RCT design have been reported accordingly throughout the updated manuscript. Consequently, we have provided a new sample size description following the literature recommendation, as following:

Pages 17-18, lines 363 to 371: “The sample size calculation was conducted using four main characteristics: significance level, power, an estimate of group difference and standard deviation.67 As there are no published studies on the effectiveness of Sports Stars intervention in children with ASD, to the ability to estimate group differences and variances is limited. It is known that one of the main goals of a pilot study is to give information on sample size for a future RCT.68 Therefore, while no sample size calculation is necessary in a pilot trial, 15-20 subjects by group are suggested to estimate effects from pilot studies and determine if enrollment is sufficient to progress to a full RCT.69 Our aim is to enroll 18 participants per group as the minimum sample size.70”

Reviewer 2, Comment 2

Otherwise, I am simply noting a few typos I saw:

p. 12, end of first paragraph typo (invertion > intervention)

p. 12, near the bottom typo (unbinding > unblinding)

Fig 1. (enrolment > enrollment)

Response: Done as requested 

REVIEWER: 3: 

Reviewer 3, Comment 1

Overall though there are not many flaws in the manuscript, I have few concerns. First and foremost {very serious} is about the ‘sample Size’ section. You stated that “due to the lack of suitable research to determine effect size, the sample size calculation was performed ---” which may be true. But even for medium effect size, according to table-2 on page 158 of Cohen’s paper “A power primer” in Psychological Bulletin, 1992, vol.:112, pp 155-159 [which is a sort of summary of the excellent book by Jacob Cohen titled ‘Statistical power analysis for the behavioral sciences’, Academic Press, 1977, New York] even for medium effect size you need n=64 per group (type-I error=0.05, power=80%). Therefore, what is you stated [“the sample size was performed considering a parallel group design, with no weighted patient preference, power of 0.81, type I error rate of 0.05, means method of analysis, and a loss up to 20% over time, a total of 36 participants will be required for the study (18 per group)] then implies that there is certainly some very serious error in your calculations. Kindly check (and modify the protocol manuscript accordingly).

Response: We acknowledge that the statistical analysis and sample size calculation were the weaknesses of our study, as stated by the editor and all the three reviewers. Thus, considering all the queries raised and the reviewer’s suggestion, we decided to reformulate our paper, where a pilot RCT design have been reported accordingly throughout the updated manuscript. Consequently, we have provided a new sample size description following the literature recommendation, as following:

Pages 17-18, lines 363 to 371: “The sample size calculation was conducted using four main characteristics: significance level, power, an estimate of group difference and standard deviation.67 As there are no published studies on the effectiveness of Sports Stars intervention in children with ASD, to the ability to estimate group differences and variances is limited. It is known that one of the main goals of a pilot study is to give information on sample size for a future RCT.68 Therefore, while no sample size calculation is necessary in a pilot trial, 15-20 subjects by group are suggested to estimate effects from pilot studies and determine if enrollment is sufficient to progress to a full RCT.69 Our aim is to enroll 18 participants per group as the minimum sample size.70”

Reviewer 3, Comment 2

Note that ‘How the required minimum sample size for this study was determined is nevertheless a very-very important question [one of the important items in CONSORT checklist, item 7a] for any type of study (clinical trial or else) which needs to be discussed in adequate details {including assumptions made at the time of estimation, power of the study, software used, etc.}’. Moreover, in ‘Strengths and Weakenesses of the Study’ section you say “The sample size was calculated to provide the appropriate statistical power to detect precise between-group differences for the primary outcomes”. I very doubtful about this statement. Please re-check. Otherwise, this small sample size is acceptable only for ‘pilot’ study (but if considered as pilot study, ‘pilot’ word should appear in title).

Response: We acknowledge that the statistical analysis and sample size calculation were the weaknesses of our study, as stated by the editor and all the three reviewers. Thus, considering all the queries raised and the reviewer’s suggestion, we decided to reformulate our paper, where a pilot RCT design have been reported accordingly throughout the updated manuscript. Consequently, we have provided a new sample size description following the literature recommendation, as following:

Pages 17-18, lines 363 to 371: “The sample size calculation was conducted using four main characteristics: significance level, power, an estimate of group difference and standard deviation.67 As there are no published studies on the effectiveness of Sports Stars intervention in children with ASD, to the ability to estimate group differences and variances is limited. It is known that one of the main goals of a pilot study is to give information on sample size for a future RCT.68 Therefore, while no sample size calculation is necessary in a pilot trial, 15-20 subjects by group are suggested to estimate effects from pilot studies and determine if enrollment is sufficient to progress to a full RCT.69 Our aim is to enroll 18 participants per group as the minimum sample size.70”

Reviewer 3, Comment 3

I even doubt if just saying that (refer to ‘Methods and Analysis - Study Design’ section) that “The results will be further reported following the Consolidated Standards of Reporting Trials (CONSORT)” is sufficient? I guess, not (because the important phrase ‘CONSORT’ appears only once [though most of the important items are discussed/included] in this manuscript). It is well-known that while reporting [findings from and even planning] ‘Clinical Trial’ one should follow CONSORT guidelines [since your article type is ‘Clinical Trial’, you are supposed to cover important CONSORT items in the report or even in the ‘Protocol’].

Response: The Consolidated Standards of Test Reports (CONSORT), will be used to ensure that all items are covered by the article and are provided in supplemental material S2.

Pages 6-7, lines 114 to 119: “This will be a prospectively registered, open, two arm, pilot RCT. This manuscript was written in accordance with the SPIRIT (Standard Protocol Items: Recommendations for Interventional Trials) guidelines33 (see Supporting information S1). Forthcoming publication of trial results will be reported according to reporting standards for pilot and feasibilities studies (i.e., Consolidated Standards of Reporting Trials - CONSORT), (see Supporting information S2).34”

Reviewer 3, Comment 4

In ‘Data Analysis’ section since you stated that “Differences between groups will be calculated using unpaired t tests”, please note that, [especially because “the sample size calculation was performed applying a probabilistic model for data for trials from our primary outcome, the Goal Attainment Scaling (GAS) endpoint”], please note that though the measures/tools used are appropriate, most of them yield data that are in [at the most] ‘ordinal’ level of measurement [and not in ratio level of measurement for sure {as the score two times higher does not indicate presence of that parameter/phenomenon as double (for example, a Visual Analogue Scales VAS score or say ‘depression’ score)}]. Then application of suitable non-parametric test(s) is/are indicated/advisable [even if distribution may be ‘Gaussian’ (also called ‘normal’)]. Agreed that there is/are no non-parametric test(s)/technique(s) available to be used as alternative in all situation(s) [suitable / most desired/applicable], but should be used whenever/wherever they are available. This [‘ordinal’ level of measurement of data of primary outcome, the Goal Attainment Scaling (GAS) endpoint] also has implications in estimation of required sample size [it is sometimes suggested to inflate the sample size by 20-25% while dealing with ‘ordinal’ level of measurement data].

Response: We acknowledge that the statistical analysis and sample size calculation were the weaknesses of our study, as stated by the editor and all the three reviewers. Thus, considering all the queries raised and the reviewer’s suggestion, we decided to reformulate our paper, where a pilot RCT design have been reported accordingly throughout the updated manuscript. Consequently, we have also updated the statistical analysis to follow the new aims and purposes of the study.

Pages 17-18, lines 363 to 371: “The sample size calculation was conducted using four main characteristics: significance level, power, an estimate of group difference and standard deviation.67 As there are no published studies on the effectiveness of Sports Stars intervention in children with ASD, to the ability to estimate group differences and variances is limited. It is known that one of the main goals of a pilot study is to give information on sample size for a future RCT.68 Therefore, while no sample size calculation is necessary in a pilot trial, 15-20 subjects by group are suggested to estimate effects from pilot studies and determine if enrollment is sufficient to progress to a full RCT.69 Our aim is to enroll 18 participants per group as the minimum sample size.70”

Reviewer 3, Comment 4

As pointed out in ‘important note’ above “This review pertains only to ‘statistical aspects’ of the study and so ‘clinical aspects’ should be assessed separately/independently [one should carefully consider/look at the clinical implications of the study]. In my opinion, to rescue this article (which is quite possible), some amount of re-vision (re-drafting) may be needed. However, please do not limit the revision only (with respect) to comments made here. The respected ‘Editor’ may consider accepting if found ‘clinical implications’ (of this study) valuable. ‘Major revision’ is recommended [assuming that the respected editor would to like to give chance of improvement of the manuscript as in my opinion, the study has potential].

Response: We thank you for all que suggestions made. The manuscript was modified and revised accordingly following the new aims and purposes, considering a pilot RCT. The text were improved substantially in order to highlight its clinical impact on the ASD population and future studies on the pediatric rehabilitation field.

---

## [Decision Letter · Decision Letter 1]

13 Jul 2023

PONE-D-22-31855R1Sports Stars Brazil in children with autism spectrum disorder: A pilot randomized controlled trial protocolPLOS ONE

Dear Dr. Souto,

Thank you for submitting your manuscript to PLOS ONE. After careful consideration, we feel that it has merit but does not fully meet PLOS ONE’s publication criteria as it currently stands. Therefore, we invite you to submit a revised version of the manuscript that addresses the points raised during the review process.

We look forward to receiving your revised manuscript.

Kind regards,

Aditya Pawar

Guest Editor

PLOS ONE

Journal Requirements:

Additional Editor Comments:

Hi,

Thank you for revising the manuscript. The reviewers have further made some suggestions and a new reviewer has also asked for some clarifications. Kindly incorporate those and send a pointwise response.

Reviewers' comments:

Reviewer's Responses to Questions

**Comments to the Author**

1. Does the manuscript provide a valid rationale for the proposed study, with clearly identified and justified research questions?

Reviewer #2: Yes

Reviewer #3: Partly

Reviewer #4: Yes

2. Is the protocol technically sound and planned in a manner that will lead to a meaningful outcome and allow testing the stated hypotheses?

Reviewer #2: Yes

Reviewer #3: Partly

Reviewer #4: Yes

3. Is the methodology feasible and described in sufficient detail to allow the work to be replicable?

Reviewer #2: Yes

Reviewer #3: Yes

Reviewer #4: Yes

4. Have the authors described where all data underlying the findings will be made available when the study is complete?

Reviewer #2: Yes

Reviewer #3: Yes

Reviewer #4: Yes

5. Is the manuscript presented in an intelligible fashion and written in standard English?

Reviewer #2: Yes

Reviewer #3: Yes

Reviewer #4: Yes

6. Review Comments to the Author

You may also provide optional suggestions and comments to authors that they might find helpful in planning their study.

Reviewer #2: I am satisfied that my comments on the original version have been sufficiently addressed.

Reviewer #3: COMMENTS: It is very good that “considering all the queries raised and the reviewer’s suggestion on earlier draft, you decided to reformulate your paper as ‘pilot randomized controlled trial protocol’. However, in my knowledge the primary purpose of publishing a research protocol [of main study/trial] is a means to allow the academic community to evaluate whether subsequent analysis and results are in line with the investigators' initial objectives. Additionally, it informs the academic community on ongoing research and may avoid duplication of work. Agreed that making study protocols publicly available has the benefit of disseminating the most contemporary ideas with respect to study design and data analysis.

Nevertheless, the purpose/intention of ‘protocol for a pilot/feasibility study/trial is not known {at least to me and may be to many others as well}. Will these learned authors may please explain the purpose of protocol for this pilot study? Agreed that now this being a ‘pilot’ study in nature, sample size is not a big issue. However, [though many things are ignored (loosely looked at / evaluated)] in case of ‘pilot studies’, methodological issues need to be very rigorous followed {like in case of clinical trial, CONSORT guidelines are to be strictly observed/followed & you may know that a separate document on CONSORT for Pilot trials is available}.

Though for most of the ‘Comments’ made earlier by me, the ‘Response’ is same, namely : We acknowledge that the statistical analysis and sample size calculation were the weaknesses of our study, as stated by the editor and all the three reviewers. Thus, considering all the queries raised and the reviewer’s suggestion, we decided to reformulate our paper, where a pilot RCT design have been reported accordingly throughout the updated manuscript. Consequently, we have also updated the statistical analysis to follow the new aims and purposes of the study. Pages 17-18, lines 363 to 371: “The sample size calculation was conducted using four main characteristics: significance level, power, an estimate of group difference and standard deviation.67 As there are no published studies on the effectiveness of Sports Stars intervention in children with ASD, to the ability to estimate group differences and variances is limited. It is known that one of the main goals of a pilot study is to give information on sample size for a future RCT.68 Therefore, while no sample size calculation is necessary in a pilot trial, 15-20 subjects by group are suggested to estimate effects from pilot studies and determine if enrollment is sufficient to progress to a full RCT.69 Our aim is to enroll 18 participants per group as the minimum sample size.

Note the following: As you said, “it is known that one of the main goals of a pilot study is to give information on sample size for a future RCT” by quoting reference number 68 is 100% true. Therefore, while no sample size calculation is necessary in a pilot trial is also agreed. Further you said “15-20 subjects by group are suggested to estimate effects from pilot studies” is not agreed because “According to document ‘CONSORT for Pilot trial’ - “Formal hypothesis testing for effectiveness (or efficacy) is not recommended. The aim of a pilot trial is not to assess effectiveness (or efficacy) and it will usually be underpowered to do this””, therefore please check the reference quoted w.r.t. this statement (reference 70).

Therefore, I do not have any specific recommendation [though only as system requirement I choose major revision]. Let the respected editor decide the future course. However, I request you to kindly note that I do not wish to re-review this paper/article again (let authors do any changes on accepting earlier & present comments).

Reviewer #4: 1. In the abstract- “Methods” section- better to write “This study will be conducted with 36 participants with ASD aged 6 to 12 years old”.

2. Please expand the abbreviations at the first time they are being mentioned.

3. I am not comfortable with the conclusions mentioned in the abstract “The results from this pilot study will inform if a full clinical trial is warranted and identify any necessary modifications required to the protocol”- I am not clear on what nature of the findings from the pilot will decide if a full clinical trial is required or not. I feel that a pilot trial is more appropriate to see what planning and preparations are required to conduct a full clinical trial. These could include barriers to acceptance of the interventions, the required sample size, feasibility of delivering the intervention etc.

4. The inclusion criteria states that children aged 6-12 years will be eligible for inclusion. Any reason for such a broad age range? The effect of intervention being tested may vary depending on the age at which it is being initiated. It would be better if the authors could plan to have an age-stratified analysis, in addition to the overall analysis.

5. The exclusion criteria states that participants with cognitive, behavioral, or clinical limitations will be excluded. Will the investigators assess the child for these limitation at the time of screening for the study OR they will go by documented evidence?

6. Regarding the consenting process- will the children or their caregivers (e.g., parents) provide informed consent? I worry that some younger children may not be able to understand the trial and what is expected from them better. Further, there would be children aged under 7 years and for them “assent” may be required. It would be good if the authors could provide more clarity on this.

7. Please expand on what will the control group children receive (Nothing or the standard/routine care?). If it’s the routine care, then please elaborate on what is being currently provided under routine care.

7. PLOS authors have the option to publish the peer review history of their article (what does this mean?). If published, this will include your full peer review and any attached files.

Reviewer #2: **Yes: **Daniel M. Smith

Reviewer #3: No

Reviewer #4: **Yes: **Ravi Prakash Upadhyay

---

## [Author Response · Author response to Decision Letter 1]

8 Aug 2023

Review

Reviewer 2, Comment 1

Reviewer #2: I am satisfied that my comments on the original version have been sufficiently addressed.

Response: Thanks again for your comments.

Reviewer 3, Comment 1

COMMENTS: It is very good that “considering all the queries raised and the reviewer’s suggestion on earlier draft, you decided to reformulate your paper as ‘pilot randomized controlled trial protocol’. However, in my knowledge the primary purpose of publishing a research protocol [of main study/trial] is a means to allow the academic community to evaluate whether subsequent analysis and results are in line with the investigators' initial objectives. Additionally, it informs the academic community on ongoing research and may avoid duplication of work. Agreed that making study protocols publicly available has the benefit of disseminating the most contemporary ideas with respect to study design and data analysis. 

Nevertheless, the purpose/intention of ‘protocol for a pilot/feasibility study/trial is not known {at least to me and may be to many others as well}. Will these learned authors may please explain the purpose of protocol for this pilot study? Agreed that now this being a ‘pilot’ study in nature, sample size is not a big issue. However, [though many things are ignored (loosely looked at / evaluated)] in case of ‘pilot studies’, methodological issues need to be very rigorous followed {like in case of clinical trial, CONSORT guidelines are to be strictly observed/followed & you may know that a separate document on CONSORT for Pilot trials is available}. 

Response: 

We are in line with you. The publication of trial protocols, including pilot and feasibility trial protocols, has been advocated as an important strategy to improve transparency in the conduct and communication of pivotal trials and pilot/feasibility trials (Eldridge et al., 2016). A pilot study protocol is an important step in main study design and study rigor. There are several reasons for advocating the publication of pilot protocols as stated by CONSORT extension to pilot and feasibility:

• The rationale of a pilot trial is to investigate areas of uncertainty about the future definitive RCT

• The primary aims and objectives of a pilot trial are therefore about feasibility, and this should guide the methodology used in the pilot trial

• Assessments or measurements to address each pilot trial objective should be the focus of data collection and analysis. This might include outcome measures likely to be used in the definitive trial but, equally, it might not

• Since the aim of a pilot trial is to assess the feasibility of proceeding to the future

definitive RCT, a decision process about how to proceed needs to be built into the

design of the pilot trial. This might involve formal progression criteria to decide

whether to proceed, to proceed with amendments, or not to proceed

• Methods used to address each pilot trial objective can be qualitative or quantitative. A mixed methods approach could result in both types of data being reported within the same paper. Equally, a process evaluation or other qualitative study can be done alongside a pilot trial and reported separately in more detail

• The number of participants in a pilot study should be based on the feasibility objectives and some rationale should be given

• Formal hypothesis testing for effectiveness (or efficacy) is not recommended. The aim of a pilot trial is not to assess effectiveness (or efficacy) and it will usually be underpowered to do this Research collaboration. Publishing the pilot protocol will enhance communication among groups of researchers interested in the same area and potential collaborations that may improve recruitment, expansion of the study and avoid doubling the research efforts.

Considered all the purposes mentioned above of a feasibility study, we have better addressed and clarify in the new version of the manuscript our aims/objectives and methods to follow the CONSORT for pilot and feasibility recommendations. Please see the updates in the new version highlight in grey.

We also inserted a sentence into the discussion that highlights the benefits of the feasibility study. (Page 21, lines 437-439.

This study presents a protocol of a feasibility RCT of Sports Stars Brazil in children with ASD compared to a waitlist control group. The main objective of the randomized pilot or feasibility testing is to assess the feasibility of drive the definitive future RCT.

We also would like to emphasize that this protocol complies with the CONSORT guidelines for randomized pilot and feasibility trials. Furthermore, following the MacIntosh study (2020) [published previously on PLOS ONE], we would like to highlight that in the new version of the manuscript we divide our aims/objectives/methods into two following the Thabane et al (2010) framework, as following:

(Page 6, lines 110-121).

This paper presents a feasibility Randomized clinical trial (RCT) protocol aiming to assess the feasibility of the Sports Stars Brazil intervention for children with ASD as articulated by Thabane et al (2010) framework. This framework encompasses that the aim of feasibility study might be linked to one or more of the following four classifications: process, resources, management and scientific. In this study, we will focus on two feasibility classifications

First, we will assess the process of feasibility that will determine the ability to enroll participants, the assessments completion rates, as well as adverse effects, satisfaction and adherence. To this purpose, a priori success criteria will be established when appropriated. Second, we will evaluated the scientific feasibility for estimating the effect size and variance of nine outcomes aligned with the fPRCs components.

Reviewer 3, Comment 2

Though for most of the ‘Comments’ made earlier by me, the ‘Response’ is same, namely: We acknowledge that the statistical analysis and sample size calculation were the weaknesses of our study, as stated by the editor and all the three reviewers. Thus, considering all the queries raised and the reviewer’s suggestion, we decided to reformulate our paper, where a pilot RCT design have been reported accordingly throughout the updated manuscript. Consequently, we have also updated the statistical analysis to follow the new aims and purposes of the study. Pages 17-18, lines 363 to 371: “The sample size calculation was conducted using four main characteristics: significance level, power, an estimate of group difference and standard deviation.67 As there are no published studies on the effectiveness of Sports Stars intervention in children with ASD, to the ability to estimate group differences and variances is limited. It is known that one of the main goals of a pilot study is to give information on sample size for a future RCT.68 Therefore, while no sample size calculation is necessary in a pilot trial, 15-20 subjects by group are suggested to estimate effects from pilot studies and determine if enrollment is sufficient to progress to a full RCT.69 Our aim is to enroll 18 participants per group as the minimum sample size.

Note the following: As you said, “it is known that one of the main goals of a pilot study is to give information on sample size for a future RCT” by quoting reference number 68 is 100% true. Therefore, while no sample size calculation is necessary in a pilot trial is also agreed. Further you said “15-20 subjects by group are suggested to estimate effects from pilot studies” is not agreed because “According to document ‘CONSORT for Pilot trial’ - “Formal hypothesis testing for effectiveness (or efficacy) is not recommended. The aim of a pilot trial is not to assess effectiveness (or efficacy) and it will usually be underpowered to do this””, therefore please check the reference quoted w.r.t. this statement (reference 70).

Therefore, I do not have any specific recommendation [though only as system requirement I choose major revision]. Let the respected editor decide the future course. However, I request you to kindly note that I do not wish to re-review this paper/article again (let authors do any changes on accepting earlier & present comments).

Response: Thanks for your comments. We are in line with you in the new version of the manuscript we have better clarify our aims/purposes/methods to follow the CONSORT for pilot and feasibility study. The sample size section was rewritten, as following (Page 18, lines 382-387):

This study is designed to investigate the feasibility of conducting a future RCT to evaluate the effectiveness of Sports Stars and to build decision-making processes to guide the execution of this larger study, particularly concerning satisfaction, adherence. Despite of no sample size calculation is necessary in a feasibility trial, 15-20 subjects by group are suggested to determine if enrollment is sufficient to progress to a full RCT. Our aim is to identify 36 potential participants to reach our target sample (18 per group).

Reviewer 4, Comment 1

In the abstract- “Methods” section- better to write “This study will be conducted with 36 participants with ASD aged 6 to 12 years old”.

Response: Modified as suggested.

Reviewer 4, Comment 2

Please expand the abbreviations at the first time they are being mentioned.

Response: We appreciate your comment, abbreviations have been expanded.

Reviewer 4, Comment 3

I am not comfortable with the conclusions mentioned in the abstract “The results from this pilot study will inform if a full clinical trial is warranted and identify any necessary modifications required to the protocol”- I am not clear on what nature of the findings from the pilot will decide if a full clinical trial is required or not. I feel that a pilot trial is more appropriate to see what planning and preparations are required to conduct a full clinical trial. These could include barriers to acceptance of the interventions, the required sample size, feasibility of delivering the intervention etc.

Response: We appreciate your comment. The conclusion of the abstract has been reformulated as suggested (page 3, lines 29-31).

The results of this feasibility study will inform which components are critical to planning and preparing a future RCT study, aiming to ensure that the RCT will be feasible, rigorous and justifiable. 

Reviewer 4, Comment 4

The inclusion criteria states that children aged 6-12 years will be eligible for inclusion. Any reason for such a broad age range? The effect of intervention being tested may vary depending on the age at which it is being initiated. It would be better if the authors could plan to have an age-stratified analysis, in addition to the overall analysis.

Response: Thank you for your comment and your suggestion. The original Sports Stars Protocol was designed for children aged 6 to 12 years with Cerebral Palsy, demonstrating positive results for all age groups (Clutterbuck, Auld, & Johnston, 2022). As stated by the other reviewers, it was necessary to reformulate the purposes of our feasibility study, since its main purposes its not to estimate effects and differences between groups. Thus, the new version of the manuscript is focusing on the feasibility aspects. The results of this pilot study will inform which components are critical to planning and preparing a future RCT study, aiming to ensure that the RCT will be feasible, rigorous and justifiable. In the future, using the full RCT data we can extrapolate the data, using other statistical analysis, to try to identify any age difference, etc. 

Clutterbuck, G. L., Auld, M. L., & Johnston, L. M. (2022). SPORTS STARS: a practitioner-led, peer-group sports intervention for ambulant, school-aged children with cerebral palsy. Parent and physiotherapist perspectives. Disability and rehabilitation, 44(6), 956-965.

Reviewer 4, Comment 5

The exclusion criteria states that participants with cognitive, behavioral, or clinical limitations will be excluded. Will the investigators assess the child for these limitation at the time of screening for the study OR they will go by documented evidence?

Response: No specific instruments will be applied to assess these limitations, only subjective assessment during screening or parent reporting. This limitation has been included in the Discussion section. Page 21, lines 455-458. 

Finally, no specific instrument will be applied to assess possible cognitive alterations of the participants. Only subjective assessment of cognitive functioning will be performed during screening or through parents' reports.

Reviewer 4, Comment 6

Regarding the consenting process- will the children or their caregivers (e.g., parents) provide informed consent? I worry that some younger children may not be able to understand the trial and what is expected from them better. Further, there would be children aged under 7 years and for them “assent” may be required. It would be good if the authors could provide more clarity on this.

Response: Parents and children signed the free and informed consent form and the free and informed assent form, respectively (This information can be found on page 20, lines 417 to 419). The term of assent has a language aimed at the child population, including numerous illustrations that facilitate the understanding of children. In addition, a researcher was available to assist in reading the terms and clarify any doubts the parents or children might have.

Reviewer 4, Comment 7

Please expand on what will the control group children receive (Nothing or the standard/routine care?). If it’s the routine care, then please elaborate on what is being currently provided under routine care.

Response: We appreciate your comment and agree that the information is not clear, and therefore, have been reformulated see page 16, lines 344-352).

Participants in the control group will receive their standard care, including maintaining existing occupational therapy and/or physical therapy intervention programs. Children with ASD in Brazil are expected to receive 1 to 2 sessions of occupational therapy (but not physical therapy) per week in public or private clinics Both interventions provide individualized treatment plans tailored to the needs of each child. Generally speaking, physical therapy provides a general exercise program that involves gross motor training, muscle strengthening, and balance and coordination training. Occupational therapy usually involves sensory integration, neuropsychomotor development and participation in occupations (activities of daily living and education)

References:

Eldridge, S. M., Chan, C. L., Campbell, M. J., Bond, C. M., Hopewell, S., Thabane, L., & Lancaster, G. A. (2016). CONSORT 2010 statement: extension to randomised pilot and feasibility trials. bmj, 355. 

MacIntosh, A., Desailly, E., Vignais, N., Vigneron, V., & Biddiss, E. (2020). A biofeedback-enhanced therapeutic exercise video game intervention for young people with cerebral palsy: A randomized single-case experimental design feasibility study. PLoS One, 15(6), e0234767.

Thabane L, Ma J, Chu R, Cheng J, Ismaila A, Rios LP, et al. A tutorial on pilot studies: the what, why and how. BMC Med Res Methodol. 2010;10: 1. pmid:20053272

---

## [Editor Report · Decision Letter 2]

31 Aug 2023

Sports Stars Brazil in children with autism spectrum disorder: A feasibility randomized controlled trial protocol.

PONE-D-22-31855R2

Dear Dr. Souto,

We’re pleased to inform you that your manuscript has been judged scientifically suitable for publication and will be formally accepted for publication once it meets all outstanding technical requirements.

Kind regards,

Aditya Pawar

Guest Editor

PLOS ONE
---

## [Editor Report · Acceptance letter]

26 Sep 2023

PONE-D-22-31855R2 

Sports Stars Brazil in children with autism spectrum disorder: A feasibility randomized controlled trial protocol. 

Dear Dr. Souto:

I'm pleased to inform you that your manuscript has been deemed suitable for publication in PLOS ONE. Congratulations! Your manuscript is now with our production department. 

Kind regards, 

on behalf of

Dr. Aditya Pawar 

Guest Editor

PLOS ONE